# Elimination of SOX2/OCT4-Associated Prostate Cancer Stem Cells Blocks Tumor Development and Enhances Therapeutic Response

**DOI:** 10.3390/cancers11091331

**Published:** 2019-09-08

**Authors:** Prasanna Kumar Vaddi, Mark A. Stamnes, Huojun Cao, Songhai Chen

**Affiliations:** 1The Department of Pharmacology, Roy J. and Lucille A. Carver College of Medicine, University of Iowa, Iowa City, IA 52242, USA; 2The Department of Molecular Physiology and Physics, Roy J. and Lucille A. Carver College of Medicine, University of Iowa, Iowa City, IA 52242, USA; 3The Department of Endodontics, College of Dentistry and Dental Clinics, University of Iowa, Iowa City, IA 52242, USA; 4The Department of Internal Medicine, Roy J. and Lucille A. Carver College of Medicine, University of Iowa, Iowa City, IA 52242, USA; 5The Holden Comprehensive Cancer Center, Roy J. and Lucille A. Carver College of Medicine, University of Iowa, Iowa City, IA 52242, USA

**Keywords:** cancer stem cells, SORE6 reporter, SOX2, OCT4, prostate cancer, suicide gene

## Abstract

SOX2 and OCT4 are key regulators of embryonic stem cell pluripotency. They are overexpressed in prostate cancers and have been associated with cancer stem cell (CSC) properties. However, reliable tools for detecting and targeting SOX2/OCT4-overexpressing cells are lacking, limiting our understanding of their roles in prostate cancer initiation, progression, and therapeutic resistance. Here, we show that a fluorescent reporter called SORE6 can identify SOX2/OCT4-overexpressing prostate cancer cells. Among tumor cells, the SORE6 reporter identified a small fraction with CSC hallmarks: rapid self-renewal, the capability to form tumors and metastasize, and resistance to chemotherapies. Transcriptome and biochemical analyses identified PI3K/AKT signaling as critical for maintaining the SORE6^+^ population. Moreover, a SORE6-driven herpes simplex virus thymidine kinase (TK) expression construct could selectively ablate SORE6^+^ cells in tumors, blocking tumor initiation and progression, and sensitizing tumors to chemotherapy. This study demonstrates a key role of SOX2/OCT4-associated prostate cancer stem cells in tumor development and therapeutic resistance, and identifies the SORE6 reporter system as a useful tool for characterizing CSCs functions in a native tumor microenvironment.

## 1. Introduction

Prostate cancer (PCa) is the most common malignancy and the second leading cause of cancer deaths in American men [1]. The progression of localized PCa to metastatic disease significantly shortens patient survival, because of the development of resistance to contemporary treatments, including the combination of castration and taxane chemotherapy [1].

One mechanism proposed to explain how PCa can progress to castration and chemotherapy resistance argues that, within a tumor, increasingly aggressive cells arise from a small population of stem-like cells—called tumor-initiating cells/cancer stem cells (CSCs) [2]. CSCs were initially discovered in acute myelogenous leukemia and later found in many types of solid tumors, including prostate cancer [3,4]. CSCs comprise the top of the tumor cell hierarchy. They are a class of pluripotent cells that share many properties with normal stem cells, including the ability to self-renew and to differentiate into various lineages. CSCs likely play a crucial role in both tumor initiation and progression and, because they are relatively resistant to many therapies including chemotherapy and radiotherapy, CSCs may be involved in therapeutic relapse. Targeting CSCs may hold promise in improving the effectiveness of cancer management [3], but specific inhibitors of CSCs are still lacking.

Nowadays, a major challenge in the CSC field is their identification, isolation and characterization [3]. There are no generic markers for CSCs, so currently they are typically identified and isolated via cell-surface or intracellular markers. In prostate cancer, multiple markers, including CD44, CD133, CD24, CD166 and aldehyde dehydrogenase (ALDH), have been used to detect and isolate CSCs in human tumor tissues or cancer cell lines [5]. These markers, however, are not specifically expressed in CSCs, and their expression levels vary in different tissues and cells, and fluctuate throughout the cell cycle. CSCs are usually characterized by their ability to regenerate tumors in mice in tumor cell transplantation assays; however, these assays do not accurately reflect cell behavior in the native microenvironment of an unperturbed tumor [3]. To better understand the functional properties of CSCs, it is critical to develop novel approaches to identify and target them.

SOX2, OCT4 and NANOG are the core triad of the master transcriptional regulators that maintain the pluripotent state of embryonic stem cells [6,7,8,9]. They have been shown to be overexpressed in CSCs and may serve as a marker for CSCs; indeed, fluorescence reporters driven by their individual promoters illuminate CSCs in various types of cancer, allowing CSCs to be marked and tracked [10]. One reporter, the SORE6 reporter, appears to be particularly useful as it provides a visual output for the activities of both SOX2 and OCT4 [11]. The SORE6 reporter was constructed using 6 tandem repeats of the SOX2 and OCT4 response element from the proximal human NANOG promoter and has been used to detect breast CSCs, but for other types of cancer its utility remains to be investigated. Recently, it has been shown that the SORE6 reporter marked a small fraction of two prostate cancer cell lines, DU145 and PC3 cells, with CSC-like properties in vitro [12,13]. However, the mechanisms underlying SOX2/OCT4 overexpression in CSCs are not fully understood, and the role of SOX2/OCT4 overexpressing cells in tumor initiation, progression and response to cancer therapies has not been well documented.

Here, we demonstrate that the SORE6 system can be used to identify and isolate SOX2/OCT4-overexpressing cells from prostate tumors. These cells exhibit with the properties we expect in prostate CSCs. A transcriptomic analysis of SORE6-positive cells showed that PI3K/AKT signaling was the key pathway needed to maintain CSCs. Moreover, we generated a suicide-gene construct, where SORE6 drives the expression of the herpes simplex virus thymidine kinase gene. This construct allows us to selectively eliminate SORE6^+^-CSCs in the bulk of tumors and provides compelling evidence that SOX2/OCT4 overexpression in prostate CSCs contributes to tumor initiation, progression, and tumor relapse from chemotherapy. Our findings should facilitate the development of specific approaches to target CSCs.

## 2. Results

### 2.1. The SORE6 Reporter Identifies a Subpopulation of Prostate Cancer Cells That Overexpress SOX2 and OCT4

To test how effectively the SORE6 reporter system identifies prostate cells overexpressing SOX2/OCT4, we transduced three prostate cancer cell lines (22Rv1, DU145 and PC3) with lentiviruses encoding the SORE6-GFP reporter, at the same multiplicity of infection (~1), and selected stably expressing lines with puromycin. As a control, we also generated cells expressing a minimal-cytomegalovirus (CMV)-promoter-driven GFP. The flow cytometry of the prostate cancer cell lines showed that SORE6-GFP was expressed by a small subpopulation of cells, constituting from 2 to 20%, depending on the cell lines and culture conditions (Figure 1A,B).

Similarly, in primary cells established from human (MDA-2a and MDA-2b) and mouse (mPCa) prostate cancers, SORE6^+^ sub-populations were identified (Figure 1B). The analysis of the SOX2 and OCT4 expression in PC3 cells showed that the SORE6^+^ cell population expressed more of these proteins than the SORE6^−^ population did (Figure 1C,D), indicating that the SORE6 reporter can identify prostate cancer cells overexpressing SOX2 and OCT4.

When the cell population identified by the SORE6 reporter was analyzed using the commonly used prostate CSC markers, CD44^+^CD133^+^ and ALDH, most were neither ALDH-positive nor CD44^+^CD133^+^ (Appendix A), but comparatively, SORE6^+^ cells contained ~3-fold more ALDH-positive cells and ~6-fold more CD44^+^CD133^+^ cells than SORE6^−^ cells (Appendix A).

### 2.2. SORE6^+^ Cells Exhibit CSC-Like Properties In Vitro and In Vivo

The CSC properties of SORE6^+^ cells were tested by tumorsphere-forming assays in vitro, to compare their self-renewal capacity to SORE6^−^ cells. Cells were sorted from PC3 and DU145 cells, and limiting-dilution analyses showed that, compared to SORE6^−^ cells, SORE6^+^ cells exhibited a ~3-fold increase in the tumorsphere-forming capacity (Figure 2A,B). Among the SORE6^+^ cells, the frequency of tumorsphere formation was calculated as 1 in 13 and 11 cells, in PC3 and DU145 lines, respectively; and, in the SORE6^−^ cells, 1 in 31 and 29 cells, respectively (Figure 2A,B). Upon serial passaging, SORE6^+^ PC3 and DU145 cells also generated comparatively more and larger tumorspheres (Figure 2C,D). Notably, the tumorspheres generated from the sorted, single SORE6^+^ cells contained both SORE6^+^ and SORE6^−^ cells, while SORE6^−^-derived tumorspheres contained only SORE6^−^ cells, suggesting that, compared to SORE6^−^ cells, SORE6^+^ are relatively undifferentiated (Figure 2C).

To verify these findings in vivo, nude mice were subcutaneously injected with increasing numbers (100, 500, 2500, and 5000) of sorted SORE6^+^ and SORE6^−^ PC3 and DU145 cells and monitored for the formation of primary tumors. Table 1; Table 2 show that, compared to SORE6^−^ cells, both SORE6^+^ PC3 and DU145 cells formed significantly more tumors in mice. The analysis of tumor sections showed that those generated from SORE6^+^ cells contained both SORE6^+^ and SORE6^−^ cells, with SORE6^+^ cells dispersed throughout the tumors as single-cells or clusters (Figure 2E). To examine the metastasis-forming capacity of SORE6+ and SORE6^−^ PC3 cells, 50,000 cells were injected into the left ventricle of nude mice, which were then monitored for the formation of metastases in multiple organs by BLI. Although none of the SORE6^−^ PC3 cells became metastatic after implantation for three months, half of the mice injected with SORE6+ cells acquired lung metastases. These findings suggest that, compared to SORE6^−^ cells, SORE6+ cells are far more prone to becoming metastatic.

### 2.3. SORE6^+^ Cells Are Relatively Resistant to Chemotherapeutics

Compared to the bulk of tumor cells, CSCs are relatively resistant to chemotherapy [2,3]. To test if SORE6^+^ cells exhibited these properties, we tested their response to docetaxel. As shown in Figure 3A,B, the docetaxel treatment, which killed over 50% of all cells, caused a 2 to 3-fold enrichment in SORE6^+^ PC3 and DU145 cells, suggesting that they are relatively more resistant to chemotherapeutics than their SORE6^−^ counterparts. In supporting this notion, the docetaxel treatment halted the growth of PC3 tumors in nude mice, but also caused the mice to generate tumors containing significantly more SORE6^+^ cells (from 10% to 25%; Figure 3C,D).

### 2.4. Increased PI3K/AKT Activities Maintain SORE6^+^ Population

To identify the signaling pathways that drive SOX2/OCT4-overexpressing CSC activities, SORE6^+^ and SORE6^−^ cells were sort-purified from PC3 cells for transcriptome profiling. At a threshold of over a 1.5-fold change and an FDA < 0.1, 38 genes scored as upregulated and 21 genes as downregulated in SORE6^+^ cells, compared to SORE6^−^ cells (Figure 4A,B). A literature search showed that many of these altered genes, such as IGF2 [14,15], FAM83A [16,17], CEACAM5 and CEACAM6 [18], STC1 [19] and MSMP [20], play a role in PCa or other types of cancer (Appendix A). A Gene Set Enrichment Analysis (GSEA) revealed the top signaling pathways enriched in SORE6^+^ cells, which included upregulated ERBB2, kras, mTOR and Wnt and downregulated PTEN signaling (Figure 4C).

Since these pathways commonly involve PI3K/AKT signaling [21,22,23,24,25], the PI3K/AKT activity was evaluated in SORE6^+^ cells. Compared to SORE6^−^ cells, AKT but not MAPK phosphorylation increased significantly in SORE6^+^ PC3 and DU145 cells (Figure 5A); however, the levels of total AKT, and the catalytic subunits of PI3K, p110α and p110β, were unchanged (Figure 5A). Notably, the treatment of PC3 and DU145 cells with the pan-PI3K and AKT inhibitors, GDC0941 and MK2206, significantly reduced the population of SORE6^+^ cells (Figure 5B–D), indicating that the upregulated PI3K/AKT pathway is critical for maintaining the SOX2/OCT4-overexpressing CSC population in PCa.

### 2.5. SORE6^+^ Cells Contribute to Tumor Initiation and Therapeutic Relapse

To determine the role of SOX2/OCT4-overexpressing CSCs in tumor initiation and recurrence, we used our SORE6 construct to express a suicide gene in PC3 cells. This lentiviral construct, SORE6-TK, expressed a suicide gene, the herpes simplex virus TK, under the control of the SORE6 promoter. TK-expressing cells are sensitive to GCV because TK converts GCV to GCV monophosphate, which is further converted to GCV triphosphate by the cancer cells’ enzymes, which in turn delays cell-cycle progression and induces cell apoptosis [26]. As expected, the treatment of PC3 cells with GCV specifically eliminated SORE6-TK^+^ cells and concordantly inhibited tumorsphere formation (Figure 6A,B), indicating the ablation of the CSC population in PC3 cells by GCV.

To examine whether the elimination of SOX2/OCT4-overexpressing CSCs affects prostate cancer formation, nude mice were subcutaneously implanted with equal numbers of PC3 cells expressing SORE6-GFP or SORE6-TK, and then treated with GCV for 21 days. Four months later, only 30% of the mice injected with SORE6-TK-expressing PC3 cells formed tumors, while 80% of the mice injected with SORE6-GFP-expressing cells generated tumors only two months later (Figure 6C). Tumor onset was also significantly delayed in mice injected with SORE6-TK-expressing PC3 cells (Figure 6C). The earliest tumor onsets were 84 days and 19 days in mice injected with SORE6-TK and SORE6-GFP, respectively (Figure 6C). Moreover, tumors grew significantly slower in mice injected with SORE6-TK-expressing PC3 cells than with SORE6-GFP-expressing cells (Figure 6D). These findings indicate that CSCs contribute to tumor initiation and progression.

To test if SOX2/OCT4-overexpressing CSCs contribute to therapeutic resistance, nude mice were implanted with SORE6-TK- or SORE6-GFP-expressing PC3 cells and maintained until they grew comparably sized tumors (~200–300 mm^3^). Then, they were treated with docetaxel (15 mg/kg, i.p., once per week) combined with GCV (25 mg/kg, i.p., daily) for three weeks (Figure 6E). The docetaxel treatment suppressed the growth of both SORE6-TK and -GFP-expressing tumors equally, but once the docetaxel treatment was stopped, the SORE6-GFP-expressing tumors resumed growth at an accelerated rate, while the growth of the SORE6-TK-expressing tumors continued to regress, over four weeks (Figure 6E). Comparing tumors treated with or without docetaxel and GCV showed that SORE6-TK^+^ cells were largely eliminated in tumors after treatment (Figure 6F). Together with our data showing that the docetaxel treatment increased SORE6^+^ populations in tumors (Figure 3D), these results indicate that CSCs contribute to chemotherapy resistance.

## 3. Discussion

### 3.1. SORE6 as a Reporter for CSCs

A growing body of evidence supports the CSC hypothesis in prostate cancer initiation and progression, therapeutic resistance and recurrence [4,27,28,29]; however, characterizing and targeting CSCs remain a challenge due to the lack of reliable tools for detecting and isolating them. SOX2 and OCT4 are overexpressed in a variety of cancers, including breast [30], prostate [31,32], lung [33] and colorectal cancers [34], and glioblastoma [35] and have been associated with CSC subpopulations in these tumors. However, approaches for detecting and targeting SOX2/OCT4-overexpressing cells remain limited. The SORE6 system was originally developed and tested in breast cancer [11]. It was recently shown to label PC3 cells that express high levels of the cancer stem cell marker CD44v6 and are resistant to doxorubicin [13], and DU145 tumorspheres grown in an agar-based non-adherent three-dimensional culture [12]. We found that it can also identify diverse populations of CSCs in different prostate cancer cells, ranging from human cancer cell lines to primary tumor cells derived from human prostate cancer or genetically engineered mouse models of prostate cancer. In supporting the activity of the SORE6 reporter as a detector of SOX2 and OCT4 expression [11], SORE6^+^ prostate cancer cells expressed higher levels of SOX2 and OCT4 than SORE6^−^ cells. The SORE6-enriched cell population partially overlapped with those detected by previously reported CSC markers, including ALDH and CD44/CD133, supporting the idea that CSCs are heterogeneous populations that may be enriched by different methods [36,37]. In supporting this notion, SORE6^−^ cells can still form tumorspheres in vitro, and the elimination of SORE6^+^ cells delayed but did not completely abolish the primary tumor formation. The SORE6 reporter system appears to be very effective in enriching CSCs with all known characteristics, including an increased tumorsphere formation in vitro and a tumor regenerative capacity in vivo, and a resistance to chemotherapies. It shall be noted that most of our studies were conducted in the prostate cancer cell lines, DU145 and PC3. Unlike a majority of human prostate cancers, these cells do not express androgen receptors. Nevertheless, our results provide support for the role of SOX2 and OCT4 in defining CSCs and the usefulness of the SORE6 reporter as a general marker to identify CSCs.

### 3.2. Involvement of PI3K/AKT Signaling in Maintaining CSCs

Through the transcriptome profiling of SORE6+ cells, we identified PI3K/AKT signaling as the pathway key to maintaining the prostate SOX2/OCT4-overexpressing CSC population. These results support previous findings of Dubrovska et al. [38,39] and indicate that targeting PI3K/AKT signaling may be an approach for overcoming therapeutic resistance by eradicating prostate CSCs. Our transcriptome analysis also identified several cancer-associated genes that are enriched in the SORE6^+^ population. Although some of these, such as IGF2 [14,15] and MSMP [20], are known to be involved in PCa development and progression, their roles in CSCs are unknown. The further characterization of these genes in CSCs may identify new markers and/or novel regulators of CSCs in PCa.

### 3.3. Targeting CSCs Specifically

Although the association of SOX2 and OCT4 expression with CSCs has been well established in diverse types of cancer, the role of these CSCs in tumor development remains to be elucidated due to the lack of specific inhibitors of CSCs. The in vivo function of CSCs is usually investigated by comparing the tumorigenicity of sorted CSCs and non-CSCs injected into mice. There are concerns that this implantation assay only reflects the clonogenic capacity of specific cancer cells rather than the actual stem cell-like properties of CSCs in an unperturbed tumor [3]. Our TK/GCV-suicide-gene system can selectively eradicate SOX2/OCT4-overexpressing CSCs in developing tumors, allowing us to evaluate how CSCs in their native tumor microenvironment contribute to tumor initiation and progression and responses to therapy. This construct should have a broader application for elucidating the functions of SOX2/OCT4-associated CSCs in multiple types of cancers.

## 4. Materials and Methods

### 4.1. Materials

Rabbit anti-AKT and anti-phospho-AKT473, anti-PI3K p110α anti-PI3K p110β, anti-ERK1/2, and anti-phospho-ERK1/2, SOX2 and OCT4 antibodies were from Cell Signaling Technology (Danvers, MA, USA). Mouse anti-actin was from Thermo Fisher Scientific (Waltham, MA, USA). Rabbit anti-Flag antibody was from Bethyl Laboratories (Montgomery, TX, USA). Human allophycocyanin-conjugated CD44 was from BD Biosciences (San Jose, CA, USA), and human phycoerythrin-conjugated CD133 was from Miltenyl Biotech (San Diego, CA, USA). The ALDEFLUOR kit was from Stemcell Technology (Vancouver, BC, Canada). GDC0980 and MK2206 were from Shelleck Chemicals (Houston, TX, USA). Docetaxel was from LC Laboratories (Woburn, MA, USA). GCV was from TCI America (Portland, OR, USA).

### 4.2. Plasmids

The lentiviral vectors for the SORE6-driven destabilized copepod GFP and mCherry expression and their corresponding control, the minimal CMV promotor-driven GFP (mCMV-GFP) and mCherry expression, were kindly provided by Dr. Lalage M. Wakefield (Laboratory of Cancer Biology and Genetics, National Cancer Institute, (Bethesda, MD, USA) [11]. The lentiviral vector for the SORE6-driven expression of the herpes simplex virus thymidine kinase gene, SORE6-TK vector, was constructed by inserting the nucleotide sequences encoding Flag-tagged TK and P2A peptide before the coding sequence for the second amino acid of the destabilized GFP. Briefly, the lentiviral vector for SORE6-GFP was digested with BstZ171 and FseI restriction enzymes and ligated with a synthesized DNA fragment encoding a portion of the SORE6-GFP backbone and Flag-tagged TK and P2A using a Gibson assembly cloning method (New England Biolabs, Ipswich, MA, USA).

### 4.3. Lentiviral Production

Lentiviruses were generated as described previously [40,41]. Lentiviruses were collected from the culture supernatants two- and three-days post-transfection and concentrated using the Lenti-X concentrator (Takara Bio, Mountain View, CA, USA).

### 4.4. Cell Culture and Establishment of Stable Cell Lines

The human prostate cancer cell lines, PC3, DU145, and 22RV1, were obtained from ATCC. The human MDA PCa 2a and 2b cell lines were kindly provided by Dr. Nora M. Navone (The University of Texas M.D. Anderson Cancer Center, Houston, TX, USA). PC3 and DU145 cells were cultured in DMEM (Invitrogen, Waltham, MA, USA) supplemented with 10% fetal bovine serum (FBS), while 22Rv1 cells were cultured in RPMI-1640 (Life Technologies, Waltham, MA, USA) supplemented with 10% FBS. MDA Pca 2a and 2b cells were cultured in BRFF-HPC1 media (Biological Research Faculty and Facility, Ijamsville, MD, USA) with 20% FBS, as described [42].

The mouse primary prostate cancer cells, PTEN^−/−^p53^−/−^, were established from prostate tumors developed from Pb-cre4-p53^fl/fl^PTEN^fl/fl^ mice [43]. The tumor was removed from a 26-week-old mouse and mechanically disaggregated into small pieces, followed by digestion with collagenase III into single cells. The cells were cultured in DMEM/F12 supplemented with 2.5% FBS, EGF (20 ng/mL), FGF (20 ng/mL), cholera toxin (100 ng/mL), insulin (5 µg/mL) and hydrocortisone (1 µg/mL). Fibroblasts were removed from adherent populations of epithelial cells during culture by using differential trypsinization. The identity of the prostate epithelial cells was confirmed by positive immunofluorescence staining of CK8 and androgen receptors, and the absence of p53 and PTEN by Western blotting.

The PC3, DU145, 22Rv1, MDA PCa 2a, MDA PCa 2b and p53^−/−^PTEN^−/−^cells were transduced with the aforementioned lentiviruses and selected with puromycin (1 µg/mL) for at least 1 week to establish stable cell lines.

### 4.5. Flow cytometry

Subconfluent cultured cells were dissociated with Accutase and resuspended in PBS for the flow cytometry analysis of the CD44 and CD133 expression [41]. The levels of the ALDH were determined in the presence or absence of the ALDH inhibitor, diethylaminobenzaldehyde (DEAB), according to the instruction of the ALDEFLUOR kit. To detect cells expressing SOX2, OCT4 or Flag-TK, dissociated cells were fixed and permeabilized with the True-Nuclear transcription factor buffer (Biolegend, San Diego, CA, USA), and then incubated with rabbit anti-SOX2 (1:100), OCT4 (1:100) or anti-Flag (1:200) antibodies. To determine the percentage of SORE6-GFP^+^ cells, tumors were digested with collagenase III, as described above. To exclude stromal cells in the tumor samples, dissociated tumor cells were first stained with biotin-labeled anti-mouse CD31, CD45 and Ter119, followed by PE/Cy7-cojugated streptavidin [41]. The samples were analyzed on an LSR II violet flow cytometer (Becton Dickson, Franklin Lakes, NJ, USA) with appropriate negative and single-color positive staining controls. For the tumor samples, CD31^−^/CD45^−^/Ter119^−^ cells were analyzed for SORE6-GFP expression. The data were analyzed using the FlowJo software (Tree Star, Ashland, OR, USA).

### 4.6. Fluorescence-Activated Cell Sorting

The cells transduced with SORE6-GFP were sorted using an Aria II cell sorter (BD Bioscience). The cells in the GFP^+^ or GFP^−^ gate were collected as SORE6^+^ and SORE6^−^ cells with over 95% purity.

### 4.7. Tumorsphere Culture

Single cells were plated on ultralow-attachment, 6-well plates for tumorsphere formation, as described previously [41]. After 7 to 14 days of culture, the tumorspheres were counted under an inverted microscope. To measure the size of the tumorspheres, images were taken at 5–10 randomly chosen areas under a phase contrast microscope and analyzed by ImageJ software. Spheres were dissociated with Accutase (Invitrogen) to generate a single cell suspension for subsequent expansion.

To quantify the tumorsphere-forming capacity of SORE6^+^ and SORE6^−^ cells in a limiting dilution assay, a series number of sorted cells (4, 8, 16, 32, 64, and 128) were plated on each row of 96-well plates coated with poly(2–hydroxyethl methacrylate). After a 7–10-day culture, the tumorspheres were imaged under a phase-contrast microscope, and their size analyzed by Image J software. The frequency of sphere formation was calculated as the fraction of wells that contained tumorspheres larger than the median size of all tumorspheres [44]. The tumorsphere-forming frequencies were calculated using an extreme limiting dilution algorithm (http://bioinf.wehi.edu.au/software/elda/) [45].

### 4.8. Xenograft Mouse Models

All animal studies were conducted in accordance with an Institutional Animal Care and Use Committee-approved protocol at the University of Iowa (The ethical code is: #7011932; approval date: 2 June 2017). Six to eight-week-old, male, nude mice were used for the studies. SORE6^+^ and SORE6^−^ PC3 or DU145 cells were sorted and implanted, by the injection of sorted cells at various numbers (100, 500, 1000 and 5000 in 100 µL PBS) either into the upper and lower flanks of nude mice or into the left ventricle (50,000 cells in 100 µL PBS). The tumor progression was monitored weekly by BLI or palpation and caliper measurement. The formation of tumors in the lung and other organs (including the brain, heart, liver, kidney, spleen, tibia and femur) was determined by post-mortem ex-vivo BLI, histological analysis by H&E staining and luciferase activity assays of tissue lysates.

To determine the effect of ganciclovir (GCV) on the initiation and progression of PC3 tumors, mice were treated with GCV (25 mg/kg, i.p., once daily) for 21 days immediately following PC3 cell implantation. To determine the effect on the tumor progression of docetaxel or docetaxel in combination with GCV, the mice were treated with docetaxel (15 mg/kg, i.p., once per week) or docetaxel plus GCV (25 mg/kg, i.p., once daily) for 3 weeks when the tumors reached a size of ~200–300 mm^3^. The tumor growth was recorded with a caliper measurement of the tumor length (L) and width (W) every 5 days. The tumor volume was calculated by the formula of length × width^2^ × 0.5.

### 4.9. Bioluminescence Imaging (BLI)

Mice were anaesthetized and injected with 100 µL D-luciferin (15 mg/mL in PBS) through retroorbital. BLI was performed using a Xenogen IVIS200 system. The bioluminescent data were analyzed via the software Living Image (Xenogen, Alameda, CA, USA) and expressed as photon flux (photons/s/cm^2^/sr) [41].

### 4.10. Cell Proliferation Assays

Cell proliferation was assessed by using XTT (2, 3–Bis (2–methoxy–4–nitro–5–sulphophenyl)–2H–tetrazolium–5–carboxanilide). The sorted SORE6^+^ and SORE6^−^ cells were seeded in 96-well plates (2000 cells/well) in a growth medium containing 10% FBS. The cell growth was assayed daily through the addition of XTT compound followed by an absorbance measurement at 450 nM in a microplate reader [41].

### 4.11. Gene Expression Analysis

The RNA was extracted from two separate pairs of sorted SORE6^+^ and SORE6^−^ PC3 cells using the RNeasy micro kit (Qiagen, Germantown, MD). The RNA quality was determined through the RNA integrity number (RIN) value using the RNA6000 assay (Agilent, Santa Clara, CA, USA). RNA with RIN > 7 was sequenced on the BGISEQ-500 platform (BGI, Shenzhen, Guangdong, China). Each sample was sequenced at a 100 bp paired-end read and a depth of 30 M reads per sample. The differential expression of SORE6^+^ and SORE6^−^ cells was analyzed using a standard RNA sequencing data analysis workflow. Briefly, (i) FastQC and MultiQC was applied to assess the read quality. Trim galore was used to remove low quality reads and adapter sequences. (ii) An alignment-free salmon was used for the transcripts quantification. Tximport was used to convert the transcript level to the gene level expression and add an annotation. (iii) DESeq2 was used to determine the differential expressed genes. An FDR value < 0.1 and Abs (fold Change) ≥ 1.5 was set as the threshold for the significantly differential expression. (IV) A gene set enrichment analysis was performed as described [46]. The RNAseq data have been deposited into GEO (GSE134428).

### 4.12. Western Blotting Analysis

Western blotting was performed using an Odyssey infrared imaging system (Li-Cor Biosciences, Lincoln, NE, USA) or chemiluminescent substrates for visualization [41].

### 4.13. Immunostaining

Resected xenograft tumors were fixed and embedded in OCT. The tissues were sectioned at 8 µm intervals, fixed with 3.8% paraformaldehyde and permeabilized with 0.5% Triton for 5 min. The tissues were stained with rabbit anti-Flag (1:100, Bethyl Laboratory, Montgomery, TX, USA), followed by Alexa488-conjugated goat anti-rabbit IgG secondary antibody (1:1000). At least 5 images per section at random fields were taken by a Leica ICC50HD microscope using a 10× lens and were analyzed by the Image J software.

### 4.14. Statistical Analysis

A statistical analysis between two or multiple groups was performed with a Student’s *t* test or ANOVA [41]. The survival curves were analyzed according to the Kaplan-Meier method [41].

## 5. Conclusions

Our studies show that the SORE6 reporter is a robust system for identifying SOX2/OCT4-overexpressing prostate cancer cells with CSC characteristics. Importantly, based on this system, we developed a valuable tool for the selective elimination of SOX2/OCT4-overexpressing CSCs, which facilitates the interrogation of their specific roles in cancer initiation and progression, and the response to therapies in a native tumor microenvironment. Our results demonstrate that SOX2/OCT4-associated CSCs play a critical role in the therapeutic resistance of prostate cancer to taxane chemotherapy. These findings suggest that SOX2 and OCT expression may have the potential to serve as a biomarker for predicting the efficacy of taxane therapy.

## Figures and Tables

**Figure 1 cancers-11-01331-f001:**
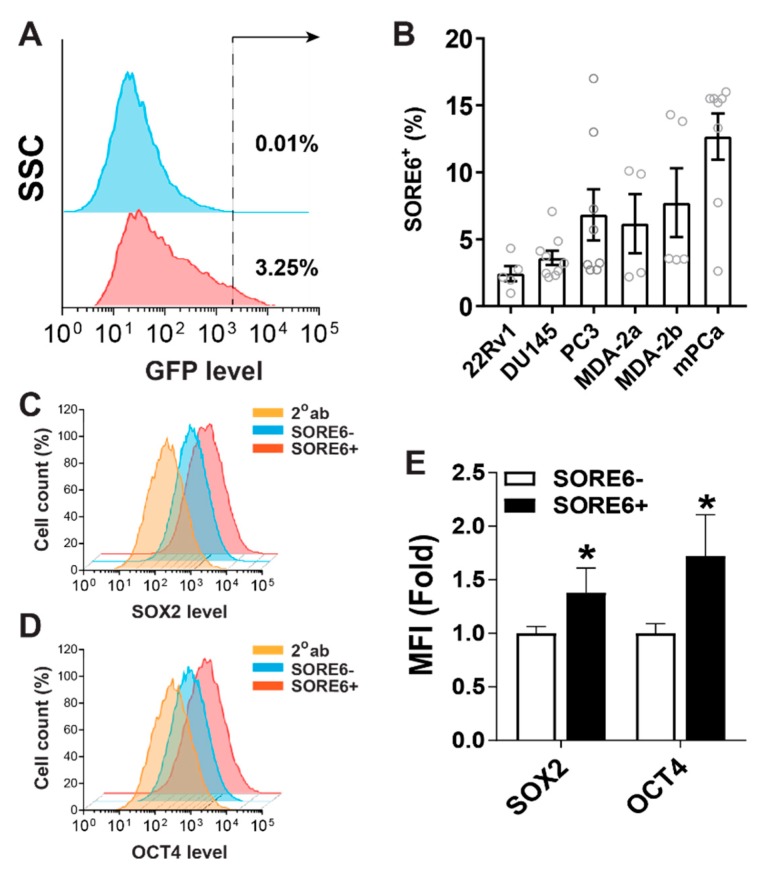
The SORE6 reporter identifies a subpopulation of PCa cells expressing high levels of SOX2 and OCT4. (**A**) A representative flow cytometry analysis of 22Rv1 cells expressing mCMV-GFP (top panel) or SORE6-GFP (bottom panel). (**B**) Percentage of SORE6^+^ cells in a panel of PCa cells identified by flow cytometry. (**C**–**E**) Flow cytometry analysis of SOX2 and OCT4 expression in SORE6^+^ and SORE6^−^ PC3 cells. The representative flow cytometry plots are shown in (**C**) (SOX2) and (**D**) (OCT4). The cells were stained with anti-SOX2 or OCT4 antibody followed by Alexa568-conjugated secondary antibody or secondary antibody alone (2°ab). The mean fluorescence intensity (MFI) from 3 to 4 independent experiments expressed as fold changes of MFI in SORE6^+^ cells over SORE6^−^cells is shown in (**E**). * *p* < 0.05 versus SORE6^−^.

**Figure 2 cancers-11-01331-f002:**
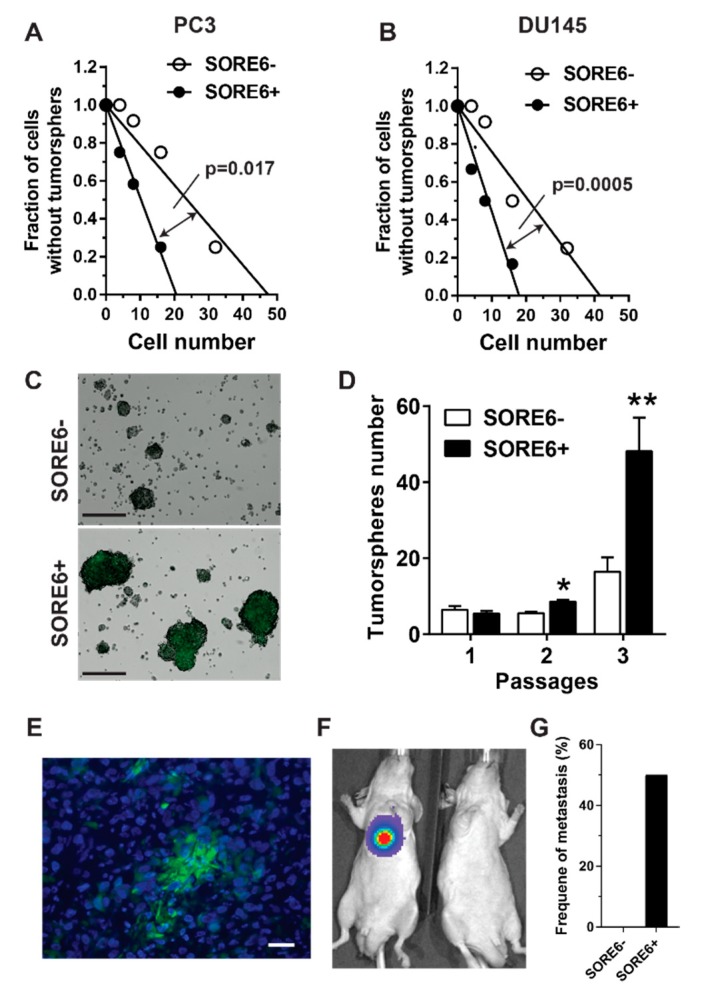
Increased self-renewal, tumorsphere- and metastasis-forming frequency in SORE6^+^ cells. SORE6^+^ and SORE6^−^ cells were sorted from (**A**,**C**–**F**) PC3 and (**B**) DU145 cells and plated on (**A**,**B**) 96-well plates for limiting dilution assays, (**C**,**D**) 6-well plates for self-renewal and tumorsphere-forming assays, or (**E**–**G**) injected into mice via the left ventricle for metastasis-forming assays. (**A**,**B**) The plots of the PC3 (**A**) and DU145 tumorsphere formation (**B**). (**C**,**D**) The representative images and quantitative data of the PC3 cell tumorsphere formation on 6-well plates at various passages, scale bar: 200 μm *, ** *p* < 0.05 and 0.01 versus SORE6^−^, respectively, *n* = 3. (**E**) A representative fluorescence image of tumor sections prepared from tumors derived from SORE6^+^ PC3 cells, scale bar: 25 μm. (**F**,**G**) The formation of lung metastases in nude mice injected with SORE6^+^ PC3 cells. (**F**) Representative bioluminescence images and (**G**) quantitative data, of lung metastases are shown.

**Figure 3 cancers-11-01331-f003:**
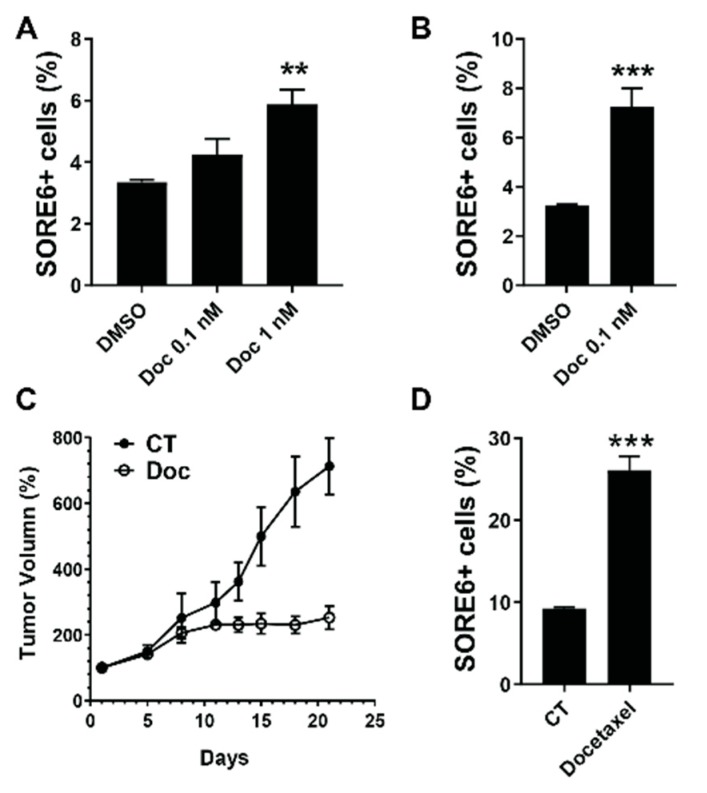
SORE6^+^ cells are relatively resistant to Docetaxel. (**A**,**B**) The effect of the docetaxel (Doc) treatment for 3 days, on the SORE6^+^ fraction in (**A**) PC3 and (**B**) DU145 cells. **, *** *p* < 0.01 and 0.001 versus DMSO, respectively, *n* = 3. (**C**,**D**) The effect of the docetaxel treatment on (**C**) primary tumor growth and (**D**) the SORE6^+^ fraction in the tumor. Nude mice (*n* = 6) were subcutaneously implanted with PC3 cells and treated with vehicle (CT) or docetaxel (15 mg/kg, i.p. once per week) for 3 weeks, once tumors reached the size of ~200–300 mm^3^. The tumor growth was measured by caliper and expressed as percentage changes of the tumor volume after the docetaxel treatment. One week after the end of the docetaxel treatment, the SORE6^+^ fraction in the tumor was determined by flow cytometry in live CD31^-^CD45^-^Ter119^−^ cells. *** *p* < 0.001 versus CT, *n* = 3–6.

**Figure 4 cancers-11-01331-f004:**
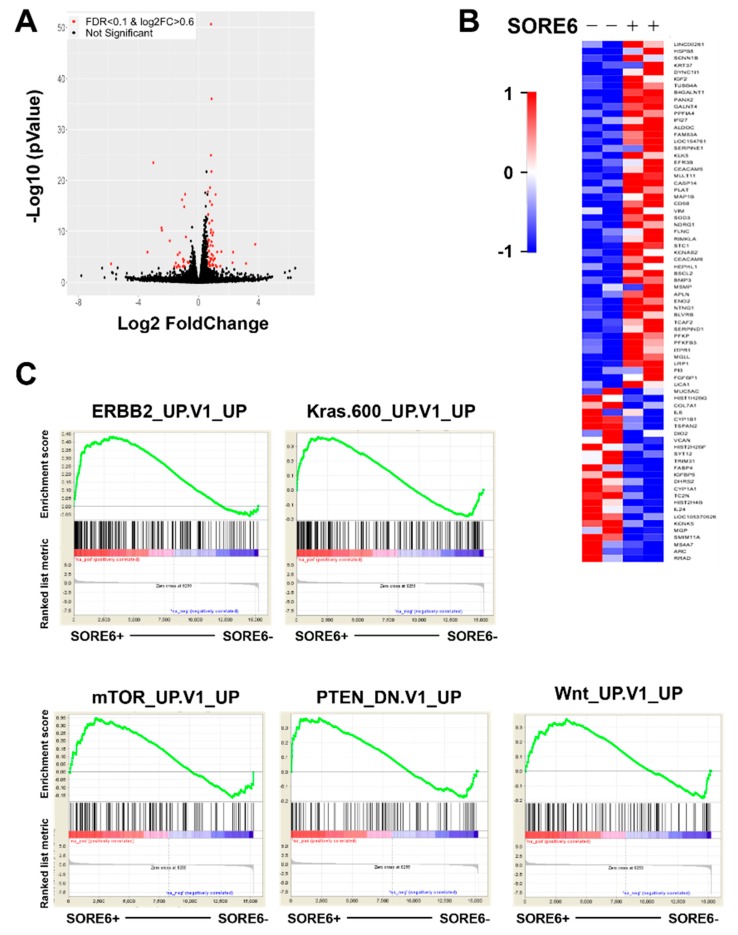
The transcriptome analysis of the altered gene expression in SORE6^+^ cells. (**A**,**B**) A volcano plot (**A**) and heatmap (**B**) show the genes differentially expressed between SORE6^+^ and SORE6^−^ PC3 cells, with a log_2_ fold change ≥ 0.6 (linear 1.5-fold change) and FDR < 0.1. The differentially expressed genes are highlighted in red in the volcano plot. (**C**) The signaling pathways significantly enriched in SORE6^+^ cells were analyzed by GSEA, with adjusted *p* < 0.05 and FDR < 0.1.

**Figure 5 cancers-11-01331-f005:**
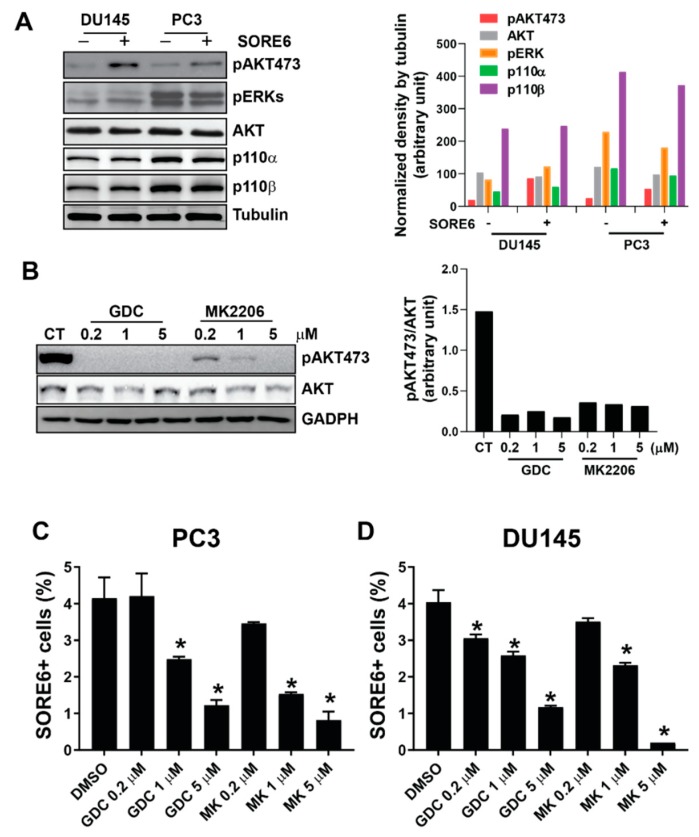
PI3K/AKT signaling is upregulated in SORE6^+^ cells. (**A**) The sorted SORE6^+^ and SORE6^−^ PC3 and DU145 cells were analyzed by Western blotting for the expression of the indicated proteins. Left panel, representative images; right panel, densitometric analysis of the indicated protein expression normalized by tubulin. (**B**) The effect of the GDC0941 (GDC) and MK2206 treatments on AKT phosphorylation. Left panel, representative images; right panel, densitometric analysis of pAKT473 levels normalized by total AKT. The original blots for Figure 5A,B can be found in Appendix A. (**C**,**D**) The effect of the GDC0941 (GDC) and MK2206 treatments on the SORE6^+^ fraction in (**C**) PC3 and (**D**) DU145. * *p* < 0.05 versus DMSO, *n* = 3–4.

**Figure 6 cancers-11-01331-f006:**
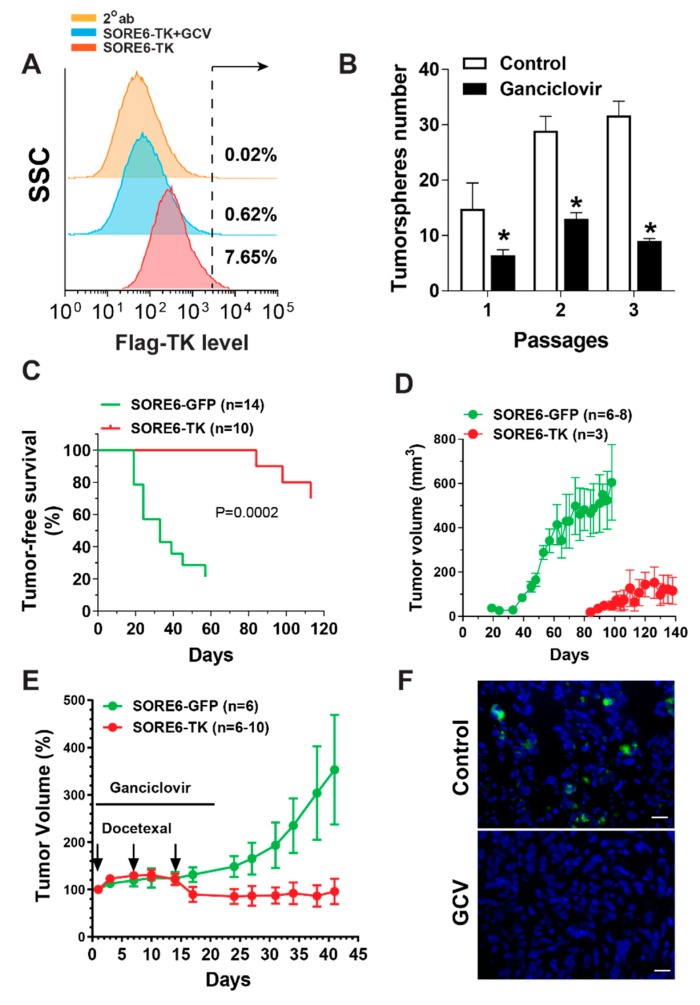
The ablation of SORE6^+^ cells blocks prostate cancer initiation and progression and enhances the therapeutic efficacy of docetaxel. (**A**) The flow cytometry analysis of SORE6-TK-expressing PC3 cells treated with ganciclovir (GCV, 10 µM) for 7 days. Cells were stained with an anti-Flag antibody followed by an Alexa488-conjugated secondary antibody or secondary antibody alone (2°ab). (**B**) The effect of GCV treatment on the tumorsphere formation of SORE6-TK-expressing PC3 cells at a series of passages. * *p* < 0.05 versus control, *n* = 3. (**C**,**E**) Nude mice were subcutaneously implanted with PC3 cells expressing SORE6-GFP or SORE6-TK, and either treated with GCV (25 mg/kg, i.p., daily) for 21 days immediately following implantation (**C**,**D**) or treated with docetaxel (15 mg/kg, i.p., once per week) in combination with GCV (25 mg/kg, i.p., daily) for 21 days when tumors reached the size of 200–300 mm^3^. The tumor growth was measured by caliper and expressed as percentage changes of the tumor volume after drug treatment. (**F**) Tumor sections were prepared from tumors derived from SORE6-TK-expressing cells and treated with or without (control) docetaxel and GCV. Sections were stained with anti-Flag antibody followed by Alexa488-conjugated secondary antibody. A representative image is shown, scale bar: 25 μm.

**Table 1 cancers-11-01331-t001:** The frequency of tumor formation in nude mice (*n* = 5) inoculated with the indicated number of SORE6^+^ and SORE6^−^ cells sorted from PC3 cells. The tumor incidence was assessed via BLI, two months after inoculation. CI, conference interval. *p*, *p* value for the difference in CSC frequencies between the SORE6^+^ and SORE6^−^ cells.

Cell Population	Number of Cells Implanted/Site				1/CSC Frequency	95% CI	*p* Value
	5000	2500	500	100			
SORE6^+^	4/5	5/5	2/5	2/5	1254	586–2683	2.17 × 10^×5^
SORE6^−^	1/5	1/5	0/5	0/5		4558–73,518	

**Table 2 cancers-11-01331-t002:** The frequency of tumor formation in nude mice (*n* = 5) inoculated with the indicated number of SORE6^+^ and SORE6^−^ cells sorted from DU145 cells. The tumor incidence was assessed via palpation, four months after inoculation. CI, conference interval. N.D., not detectable.

Cell Population	Number of Cells Implanted/Site				1/CSC Frequency	95% CI
	5000	2500	500	100		
SORE6^+^	2/5	1/5	0/5	0/5	11,279	3646–34,889
SORE6^−^	0/5	0/5	0/5	0/5	N.D.	N.D.

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
