# Peer review of "Elimination of SOX2/OCT4-Associated Prostate Cancer Stem Cells Blocks Tumor Development and Enhances Therapeutic Response"

_cancers, 2019, doi:10.3390/cancers11091331_

Round 1

Reviewer 1 Report

To the authors!

It was very interesting to read you study and I have some things that I would like to adress:

There are previous studies in prostate cancer with the SORE6 reporter system ( e.g gao et al (doi: 10.1186/s13287-018-0987-x.) and Peng et al (10.18632/oncotarget.21421) and the novel approach in this study is the sequence data, the inhibition of the PI3K/AKT pathway and the approach to introduce a suicide gene to try to eradicate the Oct4/SOX2 cells.

In the introduction and discussion I recommend that you refer to the two articles. Further you should discuss your results in relation to these previous studies.

Why you did you not test more cell lines in the respective experiments? The PC3, DU145 and 22RV1 is interestingly different and it would be great if you had all the data for all the cell lines for the different experiments. In the method (4.4) it could be understood that you did transfect all cell lines. For example it would be valuable to know the difference between PC3 and 22RV1 because the difference between the AR-expression, it would be very interesting if they differ in oct4/sox2 expression and the reaction to the docetaxel (which i belive they do) and the suicide gene, does the AR signaling in 22RV1 modify the results in any way?

Some more specific questions about the results:

Results 2.1

You present the data for SORE6 + vs – for the 22RV1 cells only, was the results similar for the other cell lines tested that you present in fig 1B.

Did you do any SOX2 and OCT4 expression flow cytometry for the other cell lines as well or only PC3? I do not think the results are so clear that you present them, for me it looks like the same number of SOX2 and OCT4 number in fig 1 c and D, or correct me if I interpret the figures wrongly. Is figure E the fold change derived from the flow cytometry from the figure 1 C and D?

Is the expression “only" between 1-2 fold change higher of oct4 and sox2 in the SORE+ sorted cells?

How did this look in the other cell lines?

What did it look like in the Sequence data in the later experiment from SORE6 sorted cells? Were OCT 2 and SOX2 differently expressed in SORE6 + vs -?

Results 2.2

If oct4/Sox2 expression is crucial for the CSC and if CSC is crucial for establishing spheres why did the SORE6 negative tumors also form spheres? Did you stain or analyze if the SORE6 negative spheres had sox2 or oct 4 expression even if theyt are SORE6 negative?

How long time did you follow the mice after injection in the left ventricle of the heart of the SORE 6 pos versus SORE6 neg cells? Did you stain for PC3 specific expression at histological examination of the lungs? According to the next experiment the time for tumor growth in SORE6 neg cells is much longer, could the results from this experiment be a lead time bias and the SORE6 tumors will eventually also give rise to metastasis?

Did you look in any other metastatic sites?

2.3

Figure 3

Fig 3A and B is incorrect text on the y-axis it should be SORE6+ instead of SORE+...:-)

I guess that the firsts reference to the figure in the figure legends should be PC3(A) and DU145(B) instead of (B) and (C). In the figure legend you refer to fig 3E... I am curious what figure it was and why you did not present that as well?

Why did you not use Doc 1nM in the DU145 experiment Fig 3B?

Other studies have shown that one mechanism of Doc resistance can be the differential expression of the ABCB1 efflux pump, were there any difference between the expression of these in the SORE6 + vs – cells ni the seq data?

2.4

You have done the experiment with PC3 cells and describe the differential expression of PTEN?!?. As I have understood the PC3 cells are PTEN -/-? (Vlietstra et al 1998 for example). Any comment on this?

2.5

In this experiment the lentiviral construct coupled to the SORE6 would make the SORE6 cells sensitive to the treatment with GCV. The results are very interesting but I would suggest that you complement the results with staining for OCT4/sox2 as well for both expermients, are oct4/sox2 stil expressed but not the SORE6 pos cells, I understand that theoretically it would not be any SORE6 negative Oct4/Sox2 pos cells, but i think it should be worth to control.

Did you test to block the PI3K/akt pathway before the thigene suicied approach? And did you try to combine the gene suicide approach with PI3K/akt pathway inhibition to see any additional effect with a triple treatment.

If the Lentivirus suicided gene model eradicate all SOX2/OCT4 cells and alls CSC are dependent of SOX2/OCT4, would the tumors not be eradicated totally?

Discussion:

I think you need to discuss previous work from other groups that also have looked at PC3 cells and the SORE6 reporter system.

I think it is also crucial to dicsuss the limitation of the PC3 model (AR-negative), clinical PC is AR-positive in the vast majority of cases.

If SORE 6 reporter system found all CSC would there be any tumor sphere formation and would there be any tumor progression at all? Fig 6 C and D? 

Author Response

Response to Reviewer 1

It was very interesting to read you study and I have some things that I would like to adress:

We would like to thank the reviewer for the thorough review of the manuscript and constructive suggestions. The following are our point-to-point responses to the comments:

There are previous studies in prostate cancer with the SORE6 reporter system (e.g gao et al (doi: 10.1186/s13287-018-0987-x.) and Peng et al (10.18632/oncotarget.21421) and the novel approach in this study is the sequence data, the inhibition of the PI3K/AKT pathway and the approach to introduce a suicide gene to try to eradicate the Oct4/SOX2 cells.

In the introduction and discussion I recommend that you refer to the two articles. Further you should discuss your results in relation to these previous studies.

Thank you for finding two previous publications using the SORE6 reporter system in DU145 and PC3M cells, respectively. As we reported, these studies showed that a subpopulation of DU145 and PC3M cells expressed the reporter. However, whether the SORE6+ cells displayed the properties of cancer stem cells, such as increased self-renewal activities in vitro and in vivo, has not been well characterized in these studies. As suggested, we have included these two papers in the reference and discussed them as appropriate.

Why you did you not test more cell lines in the respective experiments? The PC3, DU145 and 22RV1 is interestingly different and it would be great if you had all the data for all the cell lines for the different experiments. In the method (4.4) it could be understood that you did transfect all cell lines. For example it would be valuable to know the difference between PC3 and 22RV1 because the difference between the AR-expression, it would be very interesting if they differ in oct4/sox2 expression and the reaction to the docetaxel (which i belive they do) and the suicide gene, does the AR signaling in 22RV1 modify the results in any way?

We have characterized the expression of SORE6 in multiple prostate cancer cell lines. However, because of limited funding, our studies of cancer stem cell properties and their regulatory mechanisms have been focused on PC3 and DU145 cells. In future studies, as suggested, we will include other cell lines that are even more relevant to human prostate cancer, such as MDA-2a and 22Rv1.

Some more specific questions about the results:

Results 2.1

You present the data for SORE6 + vs – for the 22RV1 cells only, was the results similar for the other cell lines tested that you present in fig 1B.

Yes, only representative flow cytometric data were shown for the 22RV1 cells, but the results were similar for other cell lines. The quantitative data from several experiments done with all cell lines are shown in Fig 1B.

Did you do any SOX2 and OCT4 expression flow cytometry for the other cell lines as well or only PC3? I do not think the results are so clear that you present them, for me it looks like the same number of SOX2 and OCT4 number in fig 1 c and D, or correct me if I interpret the figures wrongly. Is figure E the fold change derived from the flow cytometry from the figure 1 C and D?

We only quantified SOX2 and OCT4 expression by flow cytometry in PC3 cells. Fig 1C and 1D showed that as compared to the secondary antibody control, SOX2 and OCT4 were expressed in both SORE6+ and SORE6- cells (as shown by the right shift of the fluorescence peak), and the levels of SOX2 and OCT4 were higher in SORE6+ cells. Fig 1E showed the fold changes of SOX2 and OCT4 calculated from the multiple experiments that are shown in Fig. 1C and 1D.

Is the expression “only" between 1-2 fold change higher of oct4 and sox2 in the SORE+ sorted cells?

How did this look in the other cell lines?

Yes, SOX2 and OCT4 only showed 1-2-fold difference in the level of expression between SORE6+ and SORE6- cells. We have not quantified SOX2 and OCT4 expression in other cell lines.

What did it look like in the Sequence data in the later experiment from SORE6 sorted cells? Were OCT 2 and SOX2 differently expressed in SORE6 + vs -?

The RNAseq data did not reveal a significant difference in the levels of SOX2 and OCT4 mRNA and we have confirmed the results by qPCR. These findings suggest that the differential expression of SOX2 and OCT4 in SORE6+ and SORE6- cells is regulated post-translationally in PC3 cells.

Results 2.2

If oct4/Sox2 expression is crucial for the CSC and if CSC is crucial for establishing spheres why did the SORE6 negative tumors also form spheres? Did you stain or analyze if the SORE6 negative spheres had sox2 or oct 4 expression even if theyt are SORE6 negative?

The reason that SORE6- cells can still form tumor spheres is probably due to the fact that they contain other populations of cancer stem cells that cannot be distinguished by SOX2 and OCT4 expression. These results are consistent with the heterogeneous nature of CSCs and our findings that the SORE6-enriched cell population partially overlapped with those detected by previously reported CSC markers, including ALDH and CD44/CD133. As we shown in Fig 1 C and 1D, SORE6- cells still expressed SOX2 and OCT4 albeit at a lower level than SORE6+ cells. We have now included these points in the Discussion (3.1).

How long time did you follow the mice after injection in the left ventricle of the heart of the SORE 6 pos versus SORE6 neg cells? Did you stain for PC3 specific expression at histological examination of the lungs? According to the next experiment the time for tumor growth in SORE6 neg cells is much longer, could the results from this experiment be a lead time bias and the SORE6 tumors will eventually also give rise to metastasis?

After injection into the left ventricle of the heart, we monitored tumor formation for up to 3 months (we have now included this information in the results). We did not perform histological examination of PC3 cells in different tissues, but we measured luciferase expression by ex vivo imaging and luciferase activity assays in different organs. Yes, it is possible that mice injected with SORE6- cells will eventually form metastases if we monitor them longer. Nonetheless, the data support the notion that cancer stem cells display enhanced metastatic potential.

Did you look in any other metastatic sites?

Yes, following bioluminescence imaging, we measured luciferase expression in multiple organs, including lung, heart, liver, kidney, brain, spleen, femur and tibia by ex vivo imaging and luciferase activity assays. We have now included this information in the methods 4.8.

2.3

Figure 3

Fig 3A and B is incorrect text on the y-axis it should be SORE6+ instead of SORE+...:-)

I guess that the firsts reference to the figure in the figure legends should be PC3(A) and DU145(B) instead of (B) and (C). In the figure legend you refer to fig 3E... I am curious what figure it was and why you did not present that as well?

Thank you for pointing out the errors in Fig 3. They are now corrected.

Why did you not use Doc 1nM in the DU145 experiment Fig 3B?

We did use 1 nM Doc in DU145 cells for Fig. 1B but found that it killed the majority of cells. Therefore flow cytometric analysis was impossible because of the low number of cells.

Other studies have shown that one mechanism of Doc resistance can be the differential expression of the ABCB1 efflux pump, were there any difference between the expression of these in the SORE6 + vs – cells ni the seq data?

We did not observe differential expression of ABCB1 or related isoforms in SORE6+ and SORE6- cells. Significantly altered genes were listed in the supplemental table 1.

2.4

You have done the experiment with PC3 cells and describe the differential expression of PTEN?!?. As I have understood the PC3 cells are PTEN -/-? (Vlietstra et al 1998 for example). Any comment on this?

No, we did not examine the differential expression of PTEN in PC3 cells. The Fig. 4 C showed that the genes overexpressed in the SORE6+ cells matched those upregulated by PTEN deficiency in a GSEA database, suggesting increased PI3K/AKT signaling.

2.5

In this experiment the lentiviral construct coupled to the SORE6 would make the SORE6 cells sensitive to the treatment with GCV. The results are very interesting but I would suggest that you complement the results with staining for OCT4/sox2 as well for both expermients, are oct4/sox2 stil expressed but not the SORE6 pos cells, I understand that theoretically it would not be any SORE6 negative Oct4/Sox2 pos cells, but i think it should be worth to control.

As we show in Fig. 1C and 1D, SORE6- cells still expressed SOX2 and OCT4 but at a lower level than SORE6+ cells. It is difficult to distinguish SOX2 and OCT4 expression between SORE6+ and SORE6- cells by immunostaining fixed tissues.

Did you test to block the PI3K/akt pathway before the thigene suicied approach? And did you try to combine the gene suicide approach with PI3K/akt pathway inhibition to see any additional effect with a triple treatment.

We have not tested the combination of PI3K/AKT inhibitors with the suicide gene approach.

If the Lentivirus suicided gene model eradicate all SOX2/OCT4 cells and alls CSC are dependent of SOX2/OCT4, would the tumors not be eradicated totally?

No, we don’t think elimination of SOX2/OCT4-expressing cells will completely eradicate primary tumors as CSCs only constitute a small population of tumor mass. However, as we showed in Fig. 6, it sensitized chemotherapy and reduced tumor recurrence, supporting a critical role of SORE6+ cells in prostate cancer.

Discussion:

I think you need to discuss previous work from other groups that also have looked at PC3 cells and the SORE6 reporter system.

I think it is also crucial to dicsuss the limitation of the PC3 model (AR-negative), clinical PC is AR-positive in the vast majority of cases.

If SORE 6 reporter system found all CSC would there be any tumor sphere formation and would there be any tumor progression at all? Fig 6 C and D?

As suggested, we have discussed previous work from other groups, the limitation of our studies using PC3 cells and the interpretation of Fig 6C and 6D (3.1).

Reviewer 2 Report

The paper by Vaddy et al. reports that a fluorescent reporter called SORE6 can identify SOX2/OCT4-overexpressing prostate cancer cells. SOX2 and OCT4 together with NANOG are considered the core triad of transcriptional regulators responsible for maintaining the pluripotent state of embryonic stem cells. Therefore, SORE6 could represent a reliable tool to detect cancer stem cells (CSC) in human tumors. In fact, the Authors report that among tumor cells, the SORE6 reporter identified a small fraction with typical CSC hallmarks such as rapid self-renewal, tumorigenicity, ability to metastasize, and drug resistance. The main messages that this paper would convey are demonstrating that SOX2/OCT overexpressing cells, as cancer stem cells, play a key role in tumor development and drug resistance and SORE6 is an efficient tool to identify such cells.

I think that this paper has some problems:

The first of them is, in my opinion, some lack of novelty. Use of SORE6 as a fluorescent reporter for SOX2/OCT4 overexpressing cells, considered as CSC, is not completely new, as it has been reported by other Authors, in many other cell types, as the Author correctly acknowledged. Also, the finding that these cells have a role in the tumorigenesis and drug resistance is somewhat expected.

As regards the experimental design, to identify the signaling pathways involved in maintaining the cells stemness, SORE6+ cells were analyzed for trascriptome profiling. this revealed 38 upregulated genes and 21 downregulated genes compared with SORE6- cells. On the basis of findings a Gene Set Enrichment analysis was used for a relatively few pathways including ERB_B2, K-Ras, mTOR and Wnt signaling. I am rather surprised that NOTCH was not investigated and would suggest of using a platform such as Ingenuity, that in my opinion could provide much more information.

Last but not least, the paper should be completely rewritten. The figure legend contain an incredibly high number of mistakes (i.e. in the table 1 and 2, 25,00 should be 2,500, I guess, the panel letters of Fig. 3 are wrong, the GSEA for PTEN and Wnt are probably inverted). Furthermore, many typos and spelling errors can be detected throughout the whole manuscript.

Author Response

Response to Reviewer 2:

The paper by Vaddy et al. reports that a fluorescent reporter called SORE6 can identify SOX2/OCT4-overexpressing prostate cancer cells. SOX2 and OCT4 together with NANOG are considered the core triad of transcriptional regulators responsible for maintaining the pluripotent state of embryonic stem cells. Therefore, SORE6 could represent a reliable tool to detect cancer stem cells (CSC) in human tumors. In fact, the Authors report that among tumor cells, the SORE6 reporter identified a small fraction with typical CSC hallmarks such as rapid self-renewal, tumorigenicity, ability to metastasize, and drug resistance. The main messages that this paper would convey are demonstrating that SOX2/OCT overexpressing cells, as cancer stem cells, play a key role in tumor development and drug resistance and SORE6 is an efficient tool to identify such cells.

I would like to thank the reviewer for helpful suggestions. The following are our point-to-point responses to the comments:

1.The first of them is, in my opinion, some lack of novelty. Use of SORE6 as a fluorescent reporter for SOX2/OCT4 overexpressing cells, considered as CSC, is not completely new, as it has been reported by other Authors, in many other cell types, as the Author correctly acknowledged. Also, the finding that these cells have a role in the tumorigenesis and drug resistance is somewhat expected.

Response: We agree that the use of SORE6 reporter to detect CSCs is not completely new. However, our studies represent the first to systematically characterize and demonstrate the usefulness of this system in detecting CSCs in prostate cancer cells. Moreover, we have elucidated the mechanisms underlying CSC regulation and developed a new tool to specifically eliminate SOX2/OCT4-associated CSCs. We believe our findings should be of great interest to Cancers readers.

2. As regards the experimental design, to identify the signaling pathways involved in maintaining the cells stemness, SORE6+ cells were analyzed for trascriptome profiling. this revealed 38 upregulated genes and 21 downregulated genes compared with SORE6- cells. On the basis of findings a Gene Set Enrichment analysis was used for a relatively few pathways including ERB_B2, K-Ras, mTOR and Wnt signaling. I am rather surprised that NOTCH was not investigated and would suggest of using a platform such as Ingenuity, that in my opinion could provide much more information.

Response: Our GSEA analysis has identified multiple pathways including NOTCH that are upregulated in SORE6+ cells. We did not present an extensive list of these pathways as they are outside the scope of this study.

3. Last but not least, the paper should be completely rewritten. The figure legend contain an incredibly high number of mistakes (i.e. in the table 1 and 2, 25,00 should be 2,500, I guess, the panel letters of Fig. 3 are wrong, the GSEA for PTEN and Wnt are probably inverted). Furthermore, many typos and spelling errors can be detected throughout the whole manuscript.

Response: Thank you for pointing out the typos and mistakes. The label for PTEN is correct, but the label for Wnt is wrong and a wrong figure is placed. They have been corrected.

Reviewer 3 Report

The manuscript entitled " Elimination of SOX2/OCT4-associated prostate cancer stem cells blocks tumor development and enhances therapeutic response" provides novel mechanistic insights into molecular and cellular determinants of therapeutic response in prostate cancer. The study focuses on  the role of SOC\X2 and OCT4, key regulators of embryonic stem cell pluripotency as markers of cancer stem cells functionally contributing to the  development of prostate cancer and the emergence of therapeutic resistance.  The authors provide critical functional in vitro and in vivo data to support their conclusions and the work is highly innovative driven by a strong mechanistic focus in the relevant translational models of prostate cancer. The findings are of major significance as the contribution of cancer stem cells in prostate tumorigenesis has not been clearly defined and the study address an unmet need.  

I have the following minor points that need to be addressed to improve the presentation of the manuscript.

1) The length of the manuscript requires shortening, specifically in the  Introduction" and "Materials and Methods" sections to eliminate detailed descriptions of standard assays and for clarity of flow.

2) To this reviewer it would be appropriate and useful for the authors to include a schematic diagram to indicate the signaling players  (SOX2/OCT4-signaling) functionally contributing to  the emergence of prostate cancer cell stemness" . Such a diagram will enable the reader to reach an appropriate interpretation of the data in the context of human prostate cancer.

3)  A statement must be included in the "Conclusions" section to describe in depth the impact of the results on therapeutic resistance to taxane chemotherapy in prostate cancer. The data from the in vivo experiments  shown on Figure 6, panels C, D and E are compelling in the context of overexpressing SOX2/OCT4 in prostate tumors dictating the therapeutic response. Will this profiling have the potential to become a signature biomarker in prostate cancer?

Author Response

Response to Reviewer 3:

Thank you for your positive review of the manuscript.

1. As suggested, we have shortened the manuscript.

2. A schematic diagram is included in the graphical abstract.

3. As suggested, we have included a statement on the potential of SOX2/OCT4 as a biomarker for taxane response in the “Conclusion” section.

Round 2

Reviewer 1 Report

Dear Authors,

I think you have commented and answered to the questions I raised about the manuscript and have now further questions.

Reviewer 2 Report

The paper has been improved. The main errors have been corrected, although I am still convinced that a more extensive investigation of the pathways involved would have been desirable.